# Antimicrobial Effect of Acetic Acid, Sodium Hypochlorite, and Thermal Treatments against Psychrotolerant *Bacillus cereus* Group Isolated from Lettuce (*Lactuca sativa* L.)

**DOI:** 10.3390/foods10092165

**Published:** 2021-09-13

**Authors:** Kyung-Min Park, Hyun-Jung Kim, Ji-Yoen Choi, Minseon Koo

**Affiliations:** 1Department of Food Analysis Center, Korea Research Institute, Wanju-gun 55365, Jeollabuk-do, Korea; Parkkyungmin@kfri.re.kr (K.-M.P.); jychoi@kfri.re.kr (J.-Y.C.); 2Department of Research Group of Consumer Safety, Korea Research Institute, Wanju-gun 55365, Jeollabuk-do, Korea; hjkim@kfri.re.kr; 3Food Biotechnology, University of Science and Technology (UST), Yuseong-gu, Daejeon 34113, Korea

**Keywords:** *Bacillus cereus* group, psychrotolerant, antimicrobials, acetic acid, sodium hypochlorite, mild heat treatment, antimicrobial activity

## Abstract

Various food products distributed throughout the cold chain can present a health risk for consumers due to the presence of psychrotolerant *B. cereus* group species that possess enterotoxin genes and antibiotic resistance. As these bacteria can grow at the low temperatures used in the food industry, this study evaluated the antimicrobial efficacy of acetic acid, sodium hypochlorite, and thermal treatments for inhibition of psychrotolerant strains and the effect that differences in activation temperature (30 °C and 10 °C) have on their efficacy. The minimum inhibitory concentration (MIC), minimum bactericidal concentration (MBC), and bacterial growth assay of acetic acid and thermal treatment showed an equal or higher antimicrobial efficacy in isolates activated at 10 °C than in those activated at 30 °C. In particular, psychrotolerant strains from the *B. cereus* group were completely eliminated with 0.25% acetic acid, regardless of the activation temperature. The possibility of tolerance was determined by observing responses in cells activated at 10 and 30 °C when exposed to different concentrations of sodium hypochlorite. Five isolates activated at 10 °C exhibited enhanced survivability in sodium hypochlorite compared to isolates activated at 30 °C, and these isolates were able to grow in sodium hypochlorite at concentrations of 250 ppm or higher. Although a significant difference in antimicrobial efficacy was observed for psychrotolerant *B. cereus* group strains depending on the activation temperature, acetic acid may be the most effective antimicrobial agent against psychrotolerant *B. cereus* species isolated from food products distributed in a cold chain.

## 1. Introduction

*Bacillus cereus*, a common soil bacterium, is one of the human pathogens associated with gastroenteritis, which is the major cause of foodborne illness in Europe [1], and is an opportunistic pathogen related to lethal non-intestinal infections in humans [2]. Some species (*B. weihenstephanensis* or *B. wiedmannii*) in the *B. cereus* group can grow at 7 °C or below; thus, cold-tolerant *B. cereus* group species are able to proliferate in refrigerated food products [3]. The *B. cereus* group comprised mostly mesophilic strains until the identification of *B. weihenstephanensis*, in 1991, as a psychrotolerant strain belonging to the *B. cereus* group [4]. One of the most recently identified psychrotolerant *B. cereus* group species, *B. wiedmannii*, is also able to grow at 7 °C or below and produces hemolysin BL (HBL) toxin and nonhemolytic enterotoxin (NHE). In addition, *B. wiedmannii* induces acute mortality and is disseminated transiently in tissues in vivo and exhibits a strong cytotoxic effect in vitro [5]. *B. wiedmannii* is negative for arginine hydrolysis and is unable to produce acid from sucrose fermentation [5]. Members of the psychrotolerant *B. cereus* group are virulent, as they produce enterotoxins or emetic toxin genes and exhibit cytotoxic activity under cold conditions [6,7]. Therefore, psychrotolerant *B. cereus* group species should be considered a food safety hazard during food processing, distribution, and storage [7].

Food processing plants use various approaches to control foodborne pathogens, including the application of antibiotics, natural or chemical antimicrobials, ionizing radiation, low temperature, and heat. Storage at low temperatures is one of various effective methods used to maintain food quality and inhibit the growth of spoilage and foodborne pathogens. However, this strategy may not be appropriate for growth inhibition of psychrotolerant bacteria since these microorganisms are able to survive at 7 °C [7,8].

The food industry applies antimicrobial agents, such as chemical or natural sanitizers, to prevent contamination by foodborne pathogens during food processing and distribution to consumers [9]. Sodium hypochlorite is the most common sanitizer used to inactivate pathogenic bacteria in food or on food contact surfaces. Many studies reported the antimicrobial activity of sodium hypochlorite in applications for food decontamination [10]. However, according to the European Commission’s expert group for technical advice on organic production, sodium hypochlorite or chlorine-based compounds are not recommended for organic farming systems [11] due to the capacity for carcinogenic product formation, such as trihalomethanes [12]. Therefore, the application of different disinfectants is required to decrease the microbial load and prevent the occurrence of foodborne outbreaks. Organic acids are being increasingly used in food safety as preservatives [13], are generally recognized as safe substances (GRAS) by the FDA, and are approved as food additives by the European Commission, FAO/WHO, and FDA [14]. Among the various organic acids, acetic acid is relatively nontoxic, inexpensive, and readily available, making it an effective and economical antimicrobial agent for eliminating foodborne pathogens. Heat treatment is a traditional microbial inactivation technique and remains a cost-effective tool to improve microbial safety [15]. However, high-temperature treatment induces substantial changes in food quality, including deterioration of the food’s nutrition, color, functionality, texture, and aroma [16]. Therefore, the application of mild heat treatment has appeared as an alternative to minimize changes in food quality [17], but unlike vegetative cells, it may be difficult to effectively inactivate bacterial spores because of their heat-resistance characteristics.

In a previous study, we isolated 18 psychrotolerant *B. cereus* group strains from lettuce, and 7 isolates were selected for further characterization as potential pathogenic isolates [7]. In general, these isolates were grown at 7 °C and showed greater biofilm formation at low temperature than at the optimal growth temperature. The psychrotolerant *B. cereus* group strains can display different characteristics against antimicrobial agents depending on the temperature. Since psychrotolerant *B. cereus* group species can grow during food transport and storage at cold temperatures, various food products distributed throughout the cold chain may present a health risk for consumers. Most relevant studies have assessed the effectiveness of antimicrobial agents against mesophilic *B. cereus* group strains. However, there is little information concerning the bactericidal efficacy against psychrotolerant strains within the *B. cereus* group. Furthermore, there are no studies on control strategies for psychrotolerant isolates at low temperatures. Therefore, the objective of this study was to determine the antimicrobial effectiveness of microbial control agents commonly used in the food processing industry (acetic acid, sodium hypochlorite, and heat) against psychrotolerant *B. cereus* group isolates. In addition, since psychrotolerant *B. cereus* group species can be exposed to temperature fluctuations, including cold temperature, the present study compared the antimicrobial activity at different culture temperatures (30 and 10 °C). Our results provide the first insight into control methods for eliminating psychrotolerant *B. cereus* group strains that are associated with foodborne illness. Understanding the response of the psychrotolerant *B. cereus* group strains to antimicrobial agents could be relevant for developing new strategies to control *B. cereus* group species, including psychrotolerant isolates, in foodstuffs.

## 2. Materials and Methods

### 2.1. Strains and Culture Conditions

Seven isolates from the *B. cereus* group (BCG) isolated from our previous study [7]. To screen the psychrotolerant strains, all isolates of the *B. cereus* group confirmed their growth ability at 42, 40, 10, 7, and 5 °C. *B. cereus* group isolates were first inoculated on tryptic soy broth (TSB, Merck, Darmstadt, Germany) and incubated at 30 °C for 18 h. Overnight culture was inoculated on a fresh TSB broth to achieve the initial concentration of 3 log colony-forming units (CFU)/mL and incubated at 42 °C for 20 h (corresponding approximately to growth temperature for mesophilic *B. cereus* group isolates), 40 °C for 20 h (corresponding approximately to maximal growth temperature for psychrotolerant *B. cereus* group isolates), 10 °C for 10 days, at 7 °C for 30 days, and at 5 °C for 45 days. Among *B. cereus* group, seven isolates (BCG34, BCG68, BCG76, BCG86, BCG88, BCG89, and BCG102) included in the study maintained the bacterial growth activity at 10 °C and 7 °C in TSB broth and showed similar lag time at 10 °C. The collected isolates did not grow at 42 °C. To prepare cells activated at 30 and 10 °C, a single colony of each isolate was first inoculated into tryptic soy broth (TSB, Merck, Darmstadt, Germany) and incubated overnight at 30 °C. One hundred microliters from each overnight culture was inoculated into fresh TSB and incubated at 10 °C for 10 days for assays conducted at 10 °C. For assays conducted at 30 °C, the cell suspensions activated at 30 °C were inoculated into fresh TSB and incubated at 30 °C for 20 h. The optical density (OD) of bacterial suspensions was measured at 600 nm using a microplate reader (Synergy Mx, Bio Tek Instruments, Inc., Vermont, VT, USA).

### 2.2. Minimum Inhibitory Concentration (MIC) and Minimum Bactericidal Concentration (MBC) Assay

To evaluate the MIC and MBC, the broth dilution method was used [18]. Acetic acid (AA, 100% solution, Sigma Aldrich, St. Louis, MO, USA) was diluted to concentrations ranging from 0.0625 to 1% (*v/v*), and sodium hypochlorite (NaOCl, 10–15% active chlorine content, Sigma Aldrich, St. Louis, MO, USA) was prepared at concentrations ranging from 50 to 250 ppm. Psychrotolerant *B. cereus* group isolates activated at 30 and 10 °C were added at a final concentration of 1 × 10^6^ colony-forming units (CFU)/mL along with acetic acid or sodium hypochlorite of specific test concentrations, as required, in 96-well plates (Falcon; Becton Dickinson Labware, Franklin Lakes, NJ, USA). Bacteria were incubated for 24 h at 30 °C, and the MIC was defined as the lowest concentration of acetic acid and sodium hypochlorite preventing visible bacterial growth. Briefly, 10 μL from each well was plated onto TSA and incubated at 30 °C for 24 h to determine whether the inhibition was reversible or permanent. The MBC was defined as the lowest concentration at which no growth was noted on TSA. The control test was performed in the same way but without acetic acid and sodium hypochlorite.

### 2.3. Bacterial Growth Analysis with Acetic Acid and Sodium Hypochlorite

To evaluate the growth activity of isolates in different concentrations, including the MBC at 30 °C, test tubes containing 5.6 mL of TSB were inoculated with 0.2 mL of the isolates activated at 30 and 10 °C, and 0.2 mL of acetic acid or sodium hypochlorite solution was added to achieve final concentrations of 0.0625, 0.125, 0.25, 0.5, 0.75, and 1% for acetic acid and 50, 100, 150, 200, and 250 ppm for sodium hypochlorite. All cultures were incubated at 30 °C and collected at 6, 12, 18, and 24 h. The collected suspensions were serially diluted in sterile 0.85% saline, and 100 μL of each dilution was inoculated onto mannitol egg yolk polymyxin agar (MYP, Merck, Darmstadt, Germany) and concentrations expressed as log CFU/mL after incubation at 30 °C for 24 h. Each experiment was duplicated and repeated using three independent trials.

### 2.4. Heat Treatment

This was performed using the modified method of Nur et al. [19], where the effects of different temperatures on bacterial growth were assessed. Briefly, 50 μL aliquots of bacterial cultures (OD_600_ = 0.1) activated at either 30 or 10 °C were inoculated into 5 mL fresh TSB and incubated at 40, 45, 50, 55, and 60 °C for 0, 10, 20, and 30 min for each temperature. Following heat treatment in the water bath, the samples were quickly transferred onto ice for cooling. Cell growth was confirmed by CFU counting. All experiments were performed in triplicate.

### 2.5. Calculation of D- and z-Values

As described by Redondo-Solano et al. [20], the *D*-value (decimal reduction times) was determined using the Excel software (2018, Microsoft) by plotting the heating time as the axis of the graph against the logarithm value of the viable cell count after heat treatment as the ordinate, where the *D*-value represents the absolute value of (1/slope) and is expressed in minutes. The thermal destruction temperature (*z*-value) was also calculated by plotting the temperature against the log *D*-value, and the data were fitted using linear regression with Excel software. The *z*-value was reported as the temperature (°C).

### 2.6. Statistical Analysis

All experiments were performed in triplicate and the results are presented as mean ± standard deviation. The results were analyzed by one-way ANOVA and Tukey’s multiple-range tests with a significance level of *p* < 0.05 using the SPSS Statistics software (IBM Corp., Armonk, NY, USA).

## 3. Results

### 3.1. Minimum Inhibitory Concentration, Minimum Bactericidal Concentration, and Effects of Acetic Acid on Bacterial Growth

The antimicrobial activity of 0.0625–1% (*v/v*) acetic acid was determined using the broth dilution method for different culture temperatures. The value of MIC against acetic acid was 0.125% for all seven isolates activated at 30 °C, and there was no difference in the MIC values (0.125%) of acetic acid for all isolates activated at 10 °C. The MBC of acetic acid for psychrotolerant *B. cereus* group isolates showed similar trends for all isolates activated at 30 and 10 °C, where the MBC was increased to 0.25% and was twofold higher than the MIC values (data not shown).

The antimicrobial activities of acetic acid against psychrotolerant *B. cereus* group isolates were determined according to the incubation time in the culture medium (i.e., TSB). Psychrotolerant *B. cereus* group isolates activated at 30 °C for 20 h and at 10 °C for 10 days were inoculated into TSB containing 0.0625, 0.125, 0.25, 0.5, and 1% acetic acid, and the number of viable cells was measured every 6 h during incubation for 24 h. Figure 1a–g shows that all isolates activated at 30 °C were completely eliminated at less than 12 h after incubation, when the final concentration was 0.25% (2× MIC, MBC) and the level of cells was maintained for 24 h. All isolates activated at 10 °C were also decreased to less than 1 log CFU/mL after 6 h with 2.5% acetic acid, and no cells were detected in any of the isolates after 24 h.

### 3.2. Minimum Inhibitory Concentration, Minimum Bactericidal Concentration, and Effects of Sodium Hypochlorite on Bacterial Growth

The MIC of sodium hypochlorite (NaOCl) was 25 ppm against all psychrotolerant *B. cereus* group isolates activated at 30 °C (Table 1), whereas isolates activated at 10 °C showed comparatively increased MIC values. When activated at 10 °C, the MIC for BCG34 and BCG68 was 50 ppm, whereas that for BCG76, BCG86, BCG88, BCG89, and BCG102 was 100 ppm.

Regardless of the activating temperature, the MBCs of NaOCl against psychrotolerant *B. cereus* group isolates were twofold higher or more than the MIC values. Furthermore, the MBCs for all isolates activated at low temperature showed higher values than those for isolates activated at optimal temperature. For instance, the MBC value increased to 300 ppm against BCG34 and BCG102 activated at 10 °C from 100 ppm for cells activated at 30 °C. Additionally, there was a two- or fourfold increase in the MBC for BCG68, BCG76, BCG86, BCG88, and BCG89 activated at 10 °C as opposed to 30 °C. Seven psychrotolerant *B. cereus* group isolates exposed at low temperature for a prolonged time exhibited higher MIC and MBC values of NaOCl than the unexposed isolates under cold conditions.

The effect of sodium hypochlorite on bacterial growth was determined using a bacterial growth assay. The assay results are shown in Figure 2, with the bacterial growth at different concentrations of NaOCl reported as the log CFU/mL in viable psychrotolerant *B. cereus* group isolates to compare the growth response of isolates activated at 30 and 10 °C. The growth of psychrotolerant *B. cereus* group isolates activated at 30 °C was inhibited at concentrations of NaOCl higher than 50 ppm. Viable cells could not be detected for any of these isolates following treatment with 200 ppm after 12 h and no regrowth within 24 h was observed, whereas some psychrotolerant *B. cereus* group isolates activated at 10 °C showed tolerance to NaOCl. When a concentration of 200 ppm was used for isolates activated at 10 °C, the growth of four isolates (BCG76, BCG86, BCG89, and BCG102) was inhibited but not completely eliminated within 24 h. In addition, while the viability of BCG34, BCG68, BCG88, BCG89, and BCG102 activated at 10 °C decreased following treatment with 250 ppm NaOCl, growth on the agar plate was observed within 24 h. In particular, when the concentration of NaOCl was 300 ppm, the viability of BCG-34 activated at 10 °C was maintained and not completely eliminated, as shown by the growth within the following 24 h.

### 3.3. Effect of Thermal Stress on the Growth of Psychrotolerant B. cereus Group Strains at Low Temperature

Seven psychrotolerant *B. cereus* group isolates were treated at temperatures between 42 and 65 °C. All psychrotolerant *B. cereus* group isolates activated at 30 °C were not significantly affected by thermal treatment. The counts of all psychrotolerant *B. cereus* group isolates activated at 30 °C were maintained their initial population (approximately 5 log CFU/mL) for one hour at all tested temperatures (42, 45, 50, 55, 60, and 65 °C), whereas the reduction rate of psychrotolerant *B. cereus* group isolates activated at 10 °C was the highest when the cells were subsequently subjected to thermal treatment at 65 °C. As shown in Table 2, the *D*-values and *z*-values obtained from the bacteria activated at low temperature indicate that some psychrotolerant *B. cereus* group isolates showed greater tolerance to thermal stress compared to other isolates, with the highest *D*-value of 6.12 min obtained for BCG-34 at 65 °C, whereas BCG89 at 65 °C had the lowest *D*-value of 2.81 min. The highest z-value of isolates activated at 10 °C was obtained for BCG34 (12.94 °C), followed by BCG102 (11.54 °C) and BCG88 (11.23 °C), whereas BCG76 showed the lowest *z*-value, at 7.72 °C.

## 4. Discussion

The consumer demand for refrigerated foods has increased due to its ability to retain the nutritional and sensorial qualities. Refrigeration is the most commonly used means of bacterial inhibition and food preservation. The procedure of cold chain and cold storage effectively controls the development of mesophilic bacterial growth. However, psychrotolerant bacteria can grow at 7 °C or lower; thus, cold storage might provide a favorable environment for the survival and growth of psychrotolerant bacteria, such as psychrotolerant *B. cereus* group strains. As the psychrotolerant *B. cereus* group strains as well as the mesophilic strains possess enterotoxin or emetic toxin genes, their existence in refrigerated food products should be monitored, and the control of these species is crucial for improving food safety. The food industry has used various approaches for bacterial inhibition, and it is desirable to have an antimicrobial agent that is rapidly applied, readily available, nontoxic, and effective.

From the broth dilution method results for acetic acid, the similar MIC (0.125%, *v/v*) and MBC (0.25%, *v/v*) values for seven psychrotolerant *B. cereus* group strains, regardless of the activation temperature, suggest that psychrotolerant *B. cereus* group isolates are vulnerable to acetic acid. Acetic acid is relatively nontoxic, inexpensive, and easily available, making it an effective and economical biocide for eliminating foodborne pathogens. A 1% concentration of acetic acid inhibits the growth of *Saccharomyces cerevisiae*, *Aspergillus niger*, *B. cereus*, and *Staphylococcus aureus* [21], and 0.15 M acetic acid (approximately 1% AA, pH 5.0) totally inhibits the growth of *B. cereus* (Wong et al., 1988). In addition, 0.4% acetic acid deactivates *Salmonella enterica* serovar Typhimurium and *Escherichia coli* cells [22], and the growth of *S. aureus* is inhibited by 0.5% acetic acid treatment [23]. *Listeria monocytogenes* exposed to 4 °C temperature showed susceptibility to lethal acid stress, such as from HCl, artificial gastric fluid, or acidified brain heart infusion broth, when compared to that grown at 37 °C [24]. Similarly, *Vibrio parahaemolyticus* exposed to a cold temperature (10 °C) showed decreased growth activity under lethal acid stress by lactic acid and acetic acid compared to cells activated at an optimal temperature [25]. Although microorganisms showed a different susceptibility to acetic acid, the growth of pathogens stopped at a less than 1% concentration of acetic acid. Low pH condition by acid stress is a common environmental threat to bacteria. The weak acid treatment can alter intracellular metabolic activities in bacteria, including protein phosphorylation, flagellar synthesis and rotation, and nutrient transport [26]. Protonated acids can enter into cells and is diffused through the cell membrane followed by intracellular dissociation of acid [27]. These changes can lead to the increase in intracellular acidity with disruption of pH homeostasis and cellular metabolism [28]. Our bacterial growth analysis results also showed that the cells of psychrotolerant *B. cereus* group strains can effectively be eliminated using a concentration of 0.25% or lower of acetic acid within 12 h, regardless of the activation temperature. These results indicate that acetic acid might be possibly used as an effective antimicrobial agent for inhibition of psychrotolerant *B. cereus* group strains in food distributed throughout cold chain.

NaOCl is used frequently as a sanitizer in the food industry and inhibits bacterial growth by disruption of enzymes, membranes, and DNA [29]. In Korea, utensils used during food processing are typically sanitized by immersion in a ‘utensil sterilizer’ containing 200 ppm sodium hypochlorite solution. In this study, the differences in the growth ability of psychrotolerant *B. cereus* group species in the presence of chlorine, activated at different temperatures, were primarily determined by measuring the MIC and MBC values for NaOCl. In isolates activated at 30 °C, the MIC of NaOCl was 25 ppm for all the isolates, and the MBC showed values increased by two- (50 ppm) or fourfold (100 ppm) compared with the MIC values. This result is consistent with previous studies showing that treatment with a 100 ppm sodium hypochlorite solution had strong inhibitory activity against species of the *B. cereus* group, with a reduction of more than 5 log CFU/mL [30]. However, low temperature exposed strains were more tolerant to sodium hypochlorite compared to the optimal temperature exposed strains. After activation at 10 °C, the MIC of NaOCl increased to 50 ppm or 100 ppm for psychrotolerant *B. cereus* group strains. The MBC for isolates activated at 10 °C was higher than the MIC. In particular, the MBC for BCG34 and BCG102 activated at 10 °C increased from 50 to 300 ppm (BCG34) and to 100 ppm (BCG102) for the MIC. The bacterial growth analysis of seven psychrotolerant *B. cereus* group isolates also showed similar trends in MIC and MBC values. In isolates cultivated at 30 °C, all isolates were completely inactivated by 200 ppm NaOCl within 12 h. However, all isolates activated at 10 °C could survive in 200 ppm NaOCl, and growth within 24 h was detected for five isolates (BCG34, BCG68, BCG88, BCG89, and BCG102) following treatment with 250 ppm NaOCl. In particular, BCG34 could grow in the presence of 300 ppm NaOCl and showed the highest growth activity in NaOCl when activated at a low temperature. It has been reported that prolonged exposure to a stress condition can enable pathogens to increase tolerance to other kinds of stress factors [31,32] and these were also confirmed in our previous [7] and present study. Other studies indicated that *L. monocytogenes* strains that were activated at low temperatures for prolonged periods exhibited cross-resistance to alkaline conditions. In *B. weihenstephanensis*, food products after refrigerated storage at 8 °C contained a high number of vegetative psychrotolerant populations at pH 3–4 [31]. Previous studies showed that the growth of psychrotolerant *B. cereus* group at low temperature enhanced their resistance to anaerobiosis [31], which increased the potential existence of stress-tolerant psychrotolerant *B. cereus* group in various food products stored at low temperatures. It has been shown that exposure to low temperature in *L. monocytogenes* induced cross-protection to high salt concentrations [33]. Pittman et al. (2014) also found that cold adapted *L. monocytogenes* to low temperature showed increased tolerance response to osmotic stress condition [8]. NaOCl is generally effective chemical antimicrobial, but psychrotolerant *B. cereus* group strains exposed at low temperature may have survival potential even after sanitizing with NaOCl.

As heat treatment is advantageous in terms of microbial inactivation, thermal treatment is still dominant in the food preservation field [34]. However, high-temperature treatment induces substantial changes in food quality, including deterioration of the food’s nutrition, color, functionality, texture, and aroma. The application of mild heat treatment has appeared as an alternative to minimize changes in the food quality and to inhibit microbial growth [17]. Furthermore, after cooking, foodstuffs are often reheated before consumption. Reheating generally involves treatment with mild heat for a long duration, which does not inhibit spore formation but may control the growth of vegetative cells or spore germination. A previous study reported that *B. cereus* group strains can survive after thermal treatment applied during food processing and can grow during distribution and storage [35,36]. However, there is insufficient information regarding the antimicrobial activity of thermal treatment against psychrotolerant *B. cereus* group strains to conclude whether heat treatment is an effective method to inhibit the growth of psychrotolerant *B. cereus* group strains. Therefore, we evaluated the effect of heat treatment against psychrotolerant *B. cereus* group strains. All psychrotolerant *B. cereus* group strains activated at 30 °C were not significantly affected by thermal treatment. The counts of all psychrotolerant *B. cereus* group strains at 30 °C were maintained for one hour at 42, 45, 50, 55, 60, and 65 °C. The growth ability of isolates activated at 10 °C under thermal treatment was lower than that of those activated at 30 °C. The reduction rate of psychrotolerant *B. cereus* group strains activated at 10 °C was the highest when the cells were subsequently subjected to thermal treatment at 65 °C. *D*-values and *z*-values obtained from the bacteria activated at low temperature indicated that some psychrotolerant *B. cereus* group strains showed comparatively higher growth ability under thermal treatment than other isolates, with the highest *D*-value and *z*-value of 6.12 min and 12.94 °C, respectively, obtained for BCG34 at 65 °C. Our finding is consistent with the results of Jackson et al. [37], who showed that cells of *E. coli* O157:H7 were more resistant to heat treatment at 37 °C than at 23 °C. Previous studies also indicated that the resistance to thermal treatment was higher when bacterial cells were activated at an optimal temperature than when these strains were grown at a relatively low temperature compared to the optimal temperature [37,38,39]. The *z*-values of psychrotolerant *B. cereus* group strains cultivated at 10 °C (between 7.27 and 14.77 °C, Table 2) were similar to or higher than those of other spore-forming bacteria and even those of non-spore-forming bacteria. For example, the *z*-value of psychrotolerant *B. cereus* vegetative cells activated at 8 °C was 5.5 °C, and the *z*-values of *S. aureus*, *S. enterica*, and *L. monocytogenes* activated at 15 °C were 6.19, 5.85, and 5.29 °C, respectively [40]. Meanwhile, spores of the *B. cereus* group strains showed higher *z*-values in comparison to vegetative cells of these strains and reported z-values in the range of 5.7 to 12 °C [35]. Den Besten et al. [41] also reported that cells of psychrotolerant *B. cereus* group strains cultivated at 12 °C were more susceptible to thermal treatment compared to cells cultivated at 30 °C. This result is in agreement with our results, since psychrotolerant *B. cereus* group strain cells activated at 10 °C were more susceptible to thermal treatment compared to cells activated at 30 °C. Mild heat treatment may be effective against psychrotolerant *B. cereus* group strains exposed at low temperature; however, this strategy may be inefficient against psychrotolerant *B. cereus* group strains activated at an optimal temperature.

## 5. Conclusions

Antimicrobials currently used in the food processing industry were examined to determine whether they were effective at eliminating psychrotolerant *B. cereus* group strains when subsequently grown at different culture temperatures. As psychrotolerant *B. cereus* group species can undergo temperature fluctuation, it is crucial to assess the efficacy of various antimicrobials against the isolates in question at the low temperatures to which they can be exposed during cold chain distribution. Exposure to a low temperature led to an increased survival ability of psychrotolerant *B. cereus* group strains exposed to sodium hypochlorite (250–300 ppm) but a similar or decreased growth response in acetic acid and after heat treatment when compared to activation at 30 °C. Since mild heat treatment did not show the antimicrobial activity in psychrotolerant strains activated at optimal temperature, acetic acid may be the most effective treatment to inhibit the growth of psychrotolerant *B. cereus* group species regardless of the activation temperature, with better antimicrobial performance than sodium hypochlorite and heat treatments.

## Figures and Tables

**Figure 1 foods-10-02165-f001:**
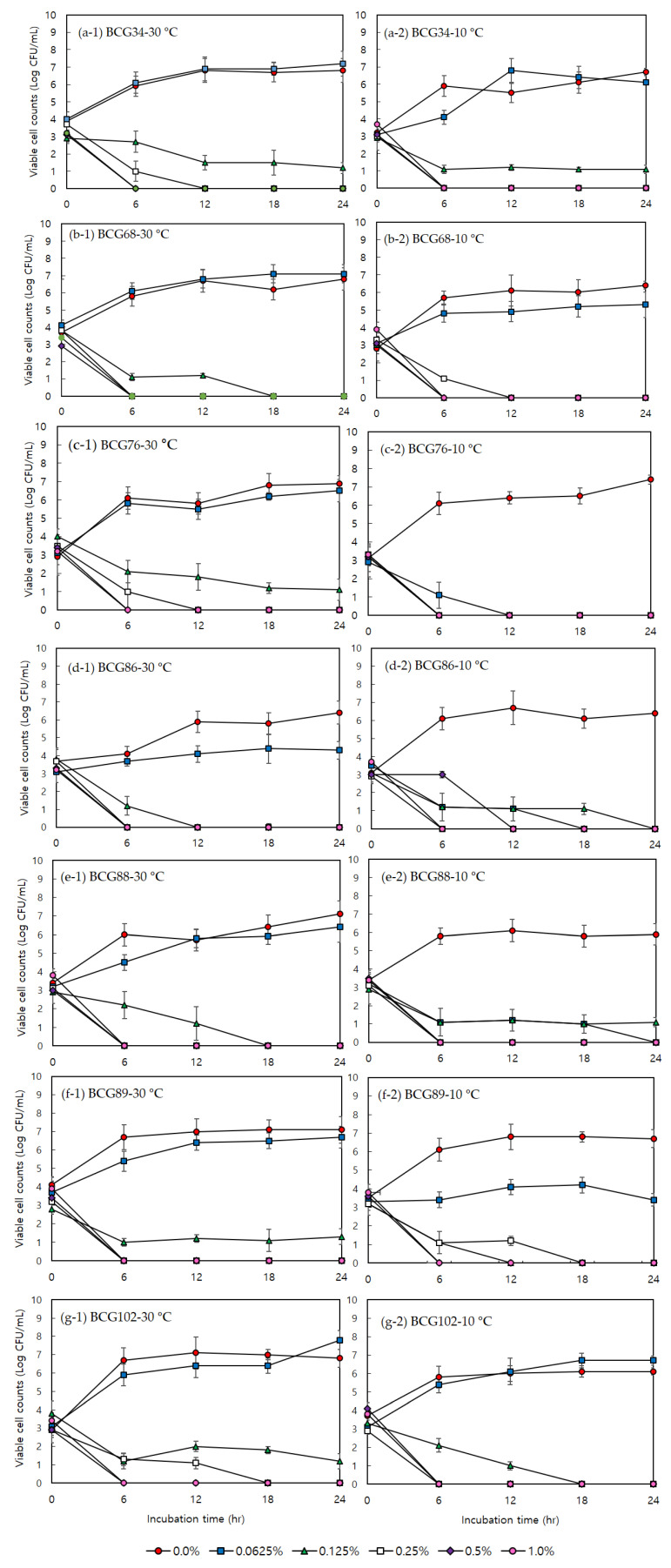
Growth curve experiments of seven psychrotolerant B. cereus group strains activated at 30 °C (**a-1**, BCG34; **b-1**, BCG68; **c-1**, BCG76; **d-1**, BCG86; **e-1**, BCG88; **f-1**, BCG89; **g-1**, BCG102) and 10 °C (**a-2**, BCG34; **b-2**, BCG68; **c-2**, BCG76; **d-2**, BCG86; **e-2**, BCG88; **f-2**, BCG89; **g-2**, BCG102) with different concentrations (0.0625, 0.125, 0.25, 0.5, and 1%) of acetic acid during 24 h. The error bars indicate standard deviation from triplicate determinations.

**Figure 2 foods-10-02165-f002:**
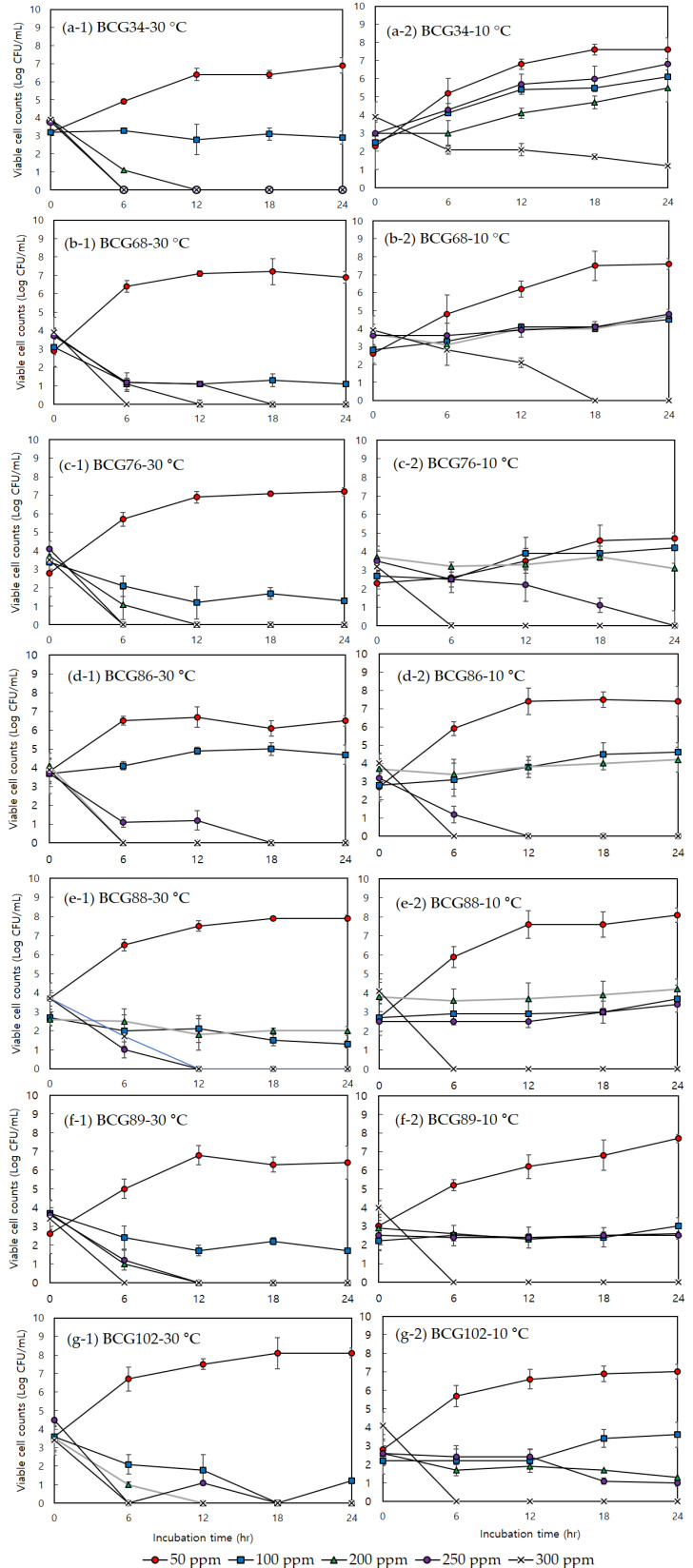
Growth curve experiments of seven psychrotolerant B. cereus group strains activated at 30 °C (**a-1**, BCG34; **b-1**, BCG68; **c-1**, BCG76; **d-1**, BCG86; **e-1**, BCG88; **f-1**, BCG89; **g-1**, BCG102) and 10 °C (**a-2**, BCG34; **b-2**, BCG68; **c-2**, BCG76; **d-2**, BCG86; **e-2**, BCG88; **f-2**, BCG89; **g-2**, BCG102) with different concentration (50, 100, 200, 250, and 300 ppm) of sodium hypochlorite during 24 hr. The error bars indicate standard deviation from triplicate determinations.

**Table 1 foods-10-02165-t001:** Minimum inhibitory concentration (MICs) and minimum bactericidal concentrations (MBCs) of sodium hypochlorite (ppm) for seven psychrotolerant *Bacillus cereus* group (BCG) strains activated at 30 °C and 10 °C.

Food Isolates	Concentration of NaOCl (ppm)			
	30°C		10 °C	
	MICs	MLCs	MICs	MLCs
BCG34	25	100	50	300
BCG68	25	50	50	200
BCG76	25	50	100	200
BCG86	25	100	100	200
BCG88	25	50	100	200
BCG89	25	100	100	200
BCG102	25	100	100	300

**Table 2 foods-10-02165-t002:** D-(min) and *z*-values (°C) of psychrotolerant Bacillus cereus group (BCG) strains activated at 10 °C at different temperatures.

Isolates	Temperature (°C)	*D*-Value * (min)	R^2^ for *D-*Value	*z-*Value * (°C)	R^2^ for *z-*Value
BCG34	55	42.2 ± 0.37	0.907	12.9 ± 0.24	0.957
	60	20.9 ± 0.17	0.909		
	65	6.1 ± 0.38	0.927		
BCG68	55	47.6 ± 0.01	0.919	9.9 ± 0.41	0.943
	60	17.9 ± 0.33	0.967		
	65	4.7 ± 0.75	0.972		
BCG76	55	66.6 ± 0.01	0.924	7.7 ± 0.39	0.949
	60	8.9 ± 0.97	0.934		
	65	3.3 ± 0.44	0.919		
BCG86	55	47.6 ± 0.01	0.912	9.4 ± 0.28	0.914
	60	9.8 ± 0.76	0.909		
	65	4.2 ± 0.99	0.917		
BCG88	55	30.3 ± 0.01	0.907	11.2 ± 0.24	0.929
	60	17.9 ± 0.37	0.924		
	65	3.9 ± 1.46	0.977		
BCG89	55	18.2 ± 0.01	0.934	10.3 ± 0.39	0.992
	60	9.8 ± 0.88	0.988		
	65	2.8 ± 4.67	0.946		
BCG102	55	27.0 ± 0.01	0.923	11.5 ± 0.17	0.948
	60	5.2 ± 1.47	0.917		
	65	3.6 ± 0.31	0.903		

* Results of triplicate determinations are expressed as mean ± standard deviation (SD).

## Data Availability

Not applicable.

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
