# Peer review of "Antimicrobial Effect of Acetic Acid, Sodium Hypochlorite, and Thermal Treatments against Psychrotolerant Bacillus cereus Group Isolated from Lettuce (Lactuca sativa L.)"

_foods, 2021, doi:10.3390/foods10092165_

Round 1
Reviewer 1 Report
A very interesting manuscript on the effect of antimicrobials against psychrotolerant B. cereus strains. The manuscript is very informative and well written. Only a few minor suggestions can be offered:
- 164-165 and Table 1. The table is not necessary and can be easily described in a couple of sentences. Please pay attention to l. 164-165 as they are not in accordance to table 1.
- 113 please keep ‘[18]’ and remove citation ‘(Andrews, 2001)’
please place B. cereus in italics (e.g. l. 172, 174, 219, 228, 234, 239, 240, 243, 280, 281, 283 etc.
- 283 please replace ‘at tested all’ with ‘at all tested’
- 298 please consider deleting ‘of foodstuffs’
- 338-339 please keep ‘[29]’ and remove citation ‘(Shikongo-Nambabi, Shoolongela, & Schneider, 2012)’
- 409. ‘Staphylococcus’ and ‘Salmonella’ can be abbreviated
- 453 please place ‘Bacillus wiedmannii’ in italics
- 502 please do not write ‘Typhimurium’ in italics
Author Response
Reviewer 1
very interesting manuscript on the effect of antimicrobials against psychrotolerant B. cereus strains. The manuscript is very informative and well written. Only a few minor suggestions can be offered:
Dear reviewer, thank you very much for your kind judgment of our manuscript. We are grateful for the time you expended to improve our work. In the following sections, you will find our responses to each of your points and suggestions.
1.164-165 and Table 1. The table is not necessary and can be easily described in a couple of sentences. Please pay attention to l. 164-165 as they are not in accordance to table 1.
Thank you for your comments. I deleted the Table 1. The results about Table 1 correctly described in sentences (174-179).
2.113 please keep ‘[18]’ and remove citation ‘(Andrews, 2001)’
I remove the author's name related to reference.
please place B. cereus in italics (e.g. l. 172, 174, 219, 228, 234, 239, 240, 243, 280, 281, 283 etc.
Thank you for your comments. I revised the B. cereus in italic in whole manuscript.
1.283 please replace ‘at tested all’ with ‘at all tested’
I revised the sentence to 'at all tested' (Line 291).
2.298 please consider deleting ‘of foodstuffs’
I deleted the words of 'of foodstuffs' (Line 306).
3.338-339 please keep ‘[29]’ and remove citation ‘(Shikongo-Nambabi, Shoolongela, & Schneider, 2012)’
I removed author's name related to reference [29] (Line 346).
4.409. ‘Staphylococcus’ and ‘Salmonella’ can be abbreviated
I revised to S. aureus and S. enterica (Line 416).
5.453 please place ‘Bacillus wiedmannii’ in italics
I revised to 'Bacillus wiedmannii' (Line 453).
6.502 please do not write ‘Typhimurium’ in italics
I revised to 'Typhimurium' from 'Typhimurium'. (Line 509)

Reviewer 2 Report
The authors have studied (with MIC and MBC) the effects of acetic acid, sodium hypochlorite and heat against seven previously isolated (by them) psychrotolerant B.cereus group species that may spoil food products or even be toxic to humans.
The authors have found that acetic acid works very well.
The B.cereus group is a very diverse group, including many species. The authors need to remind the reader how they first identified these psychrotolerant strains in their 2020 publication and whether their method of identification is accurate enough to define the exact species or just identify it as a member of the wider B.cereus group.
Minor comments:
I wonder whether the MIC and MBC for acetic acid could significantly alter the taste of the food products. A comment by the authors on this issue would be welcome.
Lines 36-38: The authors should mention in parentheses the most prominent of these B.cereus cold-tolerant species.
Line 283: (rephrase)
Line 297: …demand for refrigerated foods…
Line 317: lower case the T in typhimurium
Line 334-335: agent… inhibition of …
Line 349: … more tolerant…
Line 351: two or more spaces between the two sentences.
Line 362: tolerance
Line 372: rephrase
Line 373: antimicrobial
B. cereus needs to be italicized in some parts of the manuscript.
Author Response
Reviewer 2
The authors have studied (with MIC and MBC) the effects of acetic acid, sodium hypochlorite and heat against seven previously isolated (by them) psychrotolerant B.cereus group species that may spoil food products or even be toxic to humans.
The authors have found that acetic acid works very well.
The B.cereus group is a very diverse group, including many species. The authors need to remind the reader how they first identified these psychrotolerant strains in their 2020 publication and whether their method of identification is accurate enough to define the exact species or just identify it as a member of the wider B.cereus group.
Thank you for providing these insights about our manuscript.
I will attempt to answer you sincerely.
The B. cereus group was not traditionally considered as a psychrotolerant species until B. weihensphanenesis were identified in 1998. The term psychrotolerant is actually used for mesophilic bacteria that are low temperature tolerant and grow in a psychrophilic temperature range. Optimum growth temperature for psychrotolerant bacteria have a wider range as they can grow at 25-40°C (not grow at 42°C), and can grow at temperatures at 7°C and below. Whereas, mesophilic B. cereus group can grow at 42°C but not at less than 10°C. According to the definition (temperature criteria) for isolation of psychrotolerant strains from B. cereus group (Lotte P. Stenfors, and Per Einar Granum, 2001, FEMS Microbiology Letters, 197 (2): 223), we may perform a relatively accurate identification method for psychrotolerant B. cereus group isolates. According to your comments, I inserted the method for isolation and collection of psychrotolerant strains among B. cereus group isolated from lettuce (Line 102-113).
Minor comments:
I wonder whether the MIC and MBC for acetic acid could significantly alter the taste of the food products. A comment by the authors on this issue would be welcome.
Many organic acids are generally recognised as safe (GRAS) antimicrobials that could be used in food industry and approved to be applied in food industry for different matrices thanks to their various and useful effects. The treatment of organic acid are useful as sprays, dips, and wash additives throughout the food industry. Several studies have investigated acid washes for pathogen reduction. Household vinegar contains about 5% acetic acid and reduces the growth of pathogens in foodstuffs (Karapınar, M., and S. A. Gonul. 1992. International Journal of Food Microbiology. 16:261-264.). Acetic acid showed effective at pathogen reduction, where up to 7 log CFU/g have been shown in parsley (Wu, F. M., M. P. Doyle, L. R. Beuchat, J. G. Wells, E. D. Mintz, and B. Swaminathan. 2000. Journal of Food Protection. 63:568-572.). A 5 log reduction was shown when E. coli inoculated iceberg lettuce was treated with a diluted vinegar solution (1.9% acetic acid) (Vijayakumar, C., and C. E. Wolf-Hall. 2002. Journal of Food Protection. 65:1646-1650). Above 1% acetic acid treatment may be negatively affect sensory characteristics such as browned or wilted appearance in vegetable leaves (Vijayakumar, C., and C. E. Wolf-Hall. 2002. Journal of Food Protection. 65:1646-1650). However, the use of 0.5-1% acetic acid caused a 1 to 2 log CFU/g reduction of pathogens (E. coli and L. monocytogenes) levels on inoculated cut lettuce (Akbas, M. Y., and H. Olmez. 2007. Letters in Applied Microbiology. 44:619-624). When cut lettuce and cabbage were treated with 1% acetic, 1 log CFU/g reduction in L. monocytogenes (Zhang, S., and J. M. Farber. 1996.Food Microbiology. 13:311-321). In addition, 0.5% acetic acid showed by 3 log CFU/g in shredded lettuce against Y. enterocolitica (Escudero, M. E., L. Velazquez, M. S. Di Genaro, and A. M. S. De Guzman. 1999. Journal of Food Protection. 62:665-669). The concentration less than 0.5-1% maintained visual and sensory quality of fresh or fresh cut vegetables. In this study, the effectiveness of acetic acid for inhibition of psychrotolerant B. cereus group observed in lower concentration (MIC, 0.125%; MBC, 0.25%) than previous studies. Thus, we thought that acetic acid may be effective for controlling the microbiological safety and food quality in food products distributed throughout cold chain.
Lines 36-38: The authors should mention in parentheses the most prominent of these B.cereus cold-tolerant species.
I inserted the species name (B. weihensphanensis or B. wiedmannii) in sentence (Line 37).
Line 283: (rephrase)
I revised the sentence and insert the text of 'The counts of all pschrotolerant B. cereus group isolates activated at 30°C were maintained their initial population (approximately 5 log CFU/mL) for one hour at all tested temperatures (42, 45, 50, 55, 60, and 65 °C) (Line290).
Line 297: …demand for refrigerated foods…
I revised to ' demand for refrigerated foods' from ' demand for refrigeration of foods' (Line 305).
Line 317: lower case the T in typhimurium
According to reviewer's comments and scientific nomenclature of CDC, I revised to 'Salmonella enterica serovar Typhimurium'(Line 325).
Line 334-335: agent… inhibition of …
I corrected the sentence (Line 343).
Line 349: … more tolerant…
I corrected the words (Line 356).
Line 351: two or more spaces between the two sentences.
I deleted the space between the two sentences.
Line 362: tolerance
I revised the words (Line 380).
Line 372: rephrase
I confirmed that the text was not in accordance to reference. I revised the sentence to 'Pittman et al. (2014) also found that cold adapted L. monocytogenes to low temperature showed increased tolerance response to osmotic stress condition' (Line 379).
Line 373: antimicrobial
I revised the words (Line 381).
B. cereus needs to be italicized in some parts of the manuscript.
Thank you for your comments. I revised the B. cereus in italic in whole manuscript.
